# Competences Expected and Gained during the Teaching Practicum: Analysis of Three Competence Areas Affected during the Pandemic

Josué Prieto-Prieto [1,*], Javier Cruz-Rodríguez [2], Blanca García-Riaza [3] and María José Hernández-Serrano [4]

1 Institute of Education Sciences (IUCE), Department of Didactics of Musical, Plastic and Body Expression, University of Salamanca, 37008 Salamanca, Spain
2 Department of Didactics of Musical, Plastic and Body Expression, University of Salamanca, 37008 Salamanca, Spain; javiercruz@usal.es
3 Department of English Philology, University of Salamanca, 37008 Salamanca, Spain; bgrs@usal.es
4 Department of Theory and History of Education, University of Salamanca, 37008 Salamanca, Spain; mjhs@usal.es
* Correspondence: josueprieto@usal.es

**Abstract:** This study analyzes three competency areas promoted in the Practicum during the 2020–2021 and 2021–2022 academic years: pedagogical and didactic competences, coexistence and participation management and collaborative work. To this end, using a non-experimental design, data were collected from a sample of 725 Education students from University of Salamanca with the aims of determining the students' expectations about the Practicum prior to its development and of measuring its impact on the students during its development or the influence of the context imposed during the COVID-19 pandemic. The research was contextualized in two Practicum subjects included in the curricula of the bachelor's degrees in Early Childhood/Primary Education at the University of Salamanca. Both degrees are taught at three different university centers, Ávila, Salamanca and Zamora. The results revealed the importance of the preparation of the students in their university training period with regard to the first competence area, together with the perceptions of the students about what they learnt in the competences of areas 2 and 3. Relevant conclusions were drawn about their learning expectations towards the second area and the problems caused by the pandemic in order to develop communication skills with students and families.

**Keywords:** practicum; competences; expectations; benefits; impact; COVID-19





## 1. Introduction

Teacher training degrees in Spain, both the Early Childhood Education degree and the Primary Education degree, have a compulsory subject of curricular practices, which constitutes 20% of the credits in teacher training, as established in the law through Orders ECI/3854/2007 and ECI/3857/2007, which regulate these university studies. Since the latest reforms of the curricula, which extended the time allocated to it, Spain has become one of the European countries that devotes the most time to practical training periods [1]. Between 500 and 600 total hours are allocated for the stay in the educational centers, generally in two or three periods during different academic years. The first period constitutes the initial stage of the stay in the school center, allowing observation and familiarization with the environment, the agents and the daily reality of the classroom. In the second one, the trainees' knowledge of the educational system, the school reality and school–society relations is deepened, focusing on the understanding of the classroom and school life in its physical, social and academic dimensions [2].

These teaching, pedagogical, professional or external undergraduate curricular practices, also called Practicum, are among the most basic elements in teacher training [3,4], making it possible to bring students closer to the classroom context through professional teachers

who tutor their learning process. It is the first professional introduction to the classroom reality, to its agents and to its teaching and assessment methods. In addition, it allows the theoretical–practical contents taught in the subjects of university degrees to be combined with the professional skills necessary to practice the teaching profession in real situations experienced in a specific educational center [3]. These Practicum periods represent unique periods in which theoretical knowledge and professional practice converge, generating multiple learning experiences that must be guided through critical reflection, while trying to ensure that previous training is effectively transferred to real experience with students [4]. For many university students, these school internships become the most complex, but also the most comprehensive, subjects of the training received in their curriculum.

This relevance requires the assessment and analysis of its impact to allow us compare how and the extent to which benefits for teachers' training and professional futures are generated. In order to understand the effect of Practicum subjects on the training of future teachers, it is necessary to study the expectations towards the Practicum, as previous studies indicate that it is the most highly valued subject by students among all the formative options offered on teaching degrees, with expectations being a dimension that has scarcely been explored [5–8].

Initially, students tend to have a very positive attitude towards these subjects [9], because they know that a practical period will bring them closer to the reality of the classroom, of teachers and of students. However, recent studies suggest that it also arouses expectations and feelings of insecurity, fear or other negative emotions related to the profession [10,11]. Additional pieces of research indicate that trainees' expectations are so decisive that they allow them to rehearse their notion of their professional role [7], which may or may not be close to teachers' reality. This agrees with Peinado and Abril's observation [12] that trainees' expectations include a strong vocational idea that influences whether they reaffirm or decision to become education professionals during the Practicum.

Among the studies that have analyzed the effect of the Practicum subjects on initial ideas, attitudes, reflections, feelings or expectations, some argue that the Practicum brings both benefits and challenges. In the study by Hamaidi et al. [13], it was found that students enrolled in teacher training degrees found benefits through developing interactions and communication with students and classroom management skills, while the challenges detected included a lack of guidance from the Practicum supervisor, difficulty in communicating with teachers and a shortage of cooperation. Moussaid and Zerhouni [14] also found challenges regarding classroom control and time management. Bulgakova et al. [15] found that hurdles were associated with insecurity in feeling unprepared for professional activity. Cretu [16] also highlights that the Practicum is a mixed experience for teacher-training-degree students. Thus, the benefits include establishing a professional relationship, developing personal and professional skills and attitudes, and understanding the educational system, while challenges are associated with the implementation of teaching and the management of coexistence in the classroom.

The analysis of expectations, both positive and negative, was even more relevant in the context of the COVID-19 pandemic. Although the measures implemented in classrooms and in schools [17–19] were aimed at facilitating face-to-face teaching in a safe context, they could have affected the development of the Practicum for students on teacher training degrees. Thus, it is possible that Practicum students might have had limited opportunities to establish contact, communication and interaction with teaching staff in schools, to hold meetings with families to discuss follow-ups in tutorials or to establish links with students in schools [20–23]. On the other hand, students also appreciated positive aspects, in relation to the use of technology and the improvement of their digital competences [22,24,25], as well as regarding the possibilities offered by online learning in the educational process in any circumstance [21,23,26].

Therefore, this study focuses on determining students' expectations about the educational usefulness of this subject prior to its development, together with the perceived teacher competences acquired by the students. For this analysis, international teacher

competency frameworks, such as the European Commision Teacher Competencies framework [27,28], as well as others that have also been published in the OECD context [29–31], are used as references to select the competences or competence areas that teachers need to have in order to be effective in their work. The comparative analysis first found seven competences that could be linked to the training objectives of the Practicum subject. The seven competences were then grouped into three competence areas, as described below:

Area of Pedagogical and Didactic Competences. This refers to the ability to plan, design and carry out teaching–learning activities (the design of teaching units, the preparation of educational materials and the assessment of learning, among others). In addition, some reports include other related competences in this area: attention and adaptation to diversity and inclusion (students with special educational needs or belonging to different cultures and social contexts); digital teaching competence, which has its own entity with specific frameworks, but is also associated with the planning and development of teaching by integrating educational technologies in teaching and interaction processes; and other essential competences for teacher-training-degree students to develop the ability to reflect on their teaching practice and assess their own performance, through self-reflection and improvement competences.

Area of Coexistence and Participation Management. In this area, teachers should be able to establish a positive classroom environment to promote learning. This includes the ability to manage student behavior, encourage active participation and maintain discipline in a constructive way. In addition, it includes the competences highlighted in reports as competence for relationships and communication, which involve the acquisition of effective skills to facilitate interactions in the classroom and the educational environment, both with other teachers and with families.

Area of Competences for Collaborative Work. Teachers are expected to learn to work in teams with other teachers and professionals in the field of education. This collaboration is associated with teachers' professional development, which allows them to share ideas, experiences and resources to improve their teaching practice and enrich their students' learning.

On the basis of this analytical review and considering the regulations governing training tasks and the acquisition of competences in the Practicum in Spain for the degrees of Early Childhood Education and Primary Education, 10 competences and eight training objectives were considered for analysis in this study, which are related to the three areas of competence highlighted (see Table 1).

**Table 1.** Areas, competences and training objectives of the teacher training Practicum considered in the study.

| Areas | Competences | Objectives |
|---|---|---|
| Pedagogical and Didactic Competences, including:<br><br>- knowing how to innovate and improve.<br>- how to use educational technologies (digital competence).<br>- how to cater for diversity and promote inclusion. | (3) Learn and use different didactic strategies for the development of teaching-and-learning processes.<br>(6) Participate in the improvement proposals and the different activities proposed by an educational center, beyond the content teaching in the different areas that can be established in a center.<br>(10) Program, direct, execute and assess, with the appropriate supervision, a Teaching Unit and the student activities the teacher-tutor considers appropriate. | (e) Develop and diversify my teaching methods and strategies.<br>(f) Learn how to plan and arrange teaching.<br>(g) Prepare educational software to support teaching. |

**Table 1.** *Cont.*

| Areas | Competences | Objectives |
|---|---|---|
| Competences for the management of coexistence and participation in the classroom, including:<br><br>- communicative and interactional competence | (1) Acquire practical knowledge of the classroom and classroom management.<br>(2) Develop social and communicative skills to turn the classroom space into a place of learning and coexistence.<br>(7) To regulate the processes of interaction and communication in groups of students aged 6–12 years (Primary Education/Childhood Education) or 3–6 years (Early Childhood/Infant Education).<br>(9) Understand and establish contact and relationships with families. | (a) Develop my interaction and communication skills with pupils and families.<br>(b) Acquire skills to manage/manage the classroom.<br>(c) Increase my knowledge of the functioning and management of the school. |
| Competences for collaborative work, including:<br><br>- self-reflection-based<br>- professional development | (4) Relate the theoretical and practical concepts addressed in the different subjects of the degree with the reality of a classroom and educational center.<br>(5) Identify teachers' roles and develop them.<br>(8) Know ways of collaborating with the different sectors of the educational community and the social environment. | (d) Develop my communication and cooperation skills with my fellow teachers.<br>(h) Reaffirm my decision and vocation to become a teacher. |

Source: The Authors.

Therefore, this study is focused on establishing whether the classroom experiences of university students were in line with their initial expectations of achieving of different learning objectives. Additionally, this study analyzes the influence of the context imposed during the pandemic by COVID-19 and the limitations of social distancing, which could have affected both the trainees' expectations and their perceptions of their achievement of the competencies required in the subject.

## 2. Materials and Methods

### 2.1. Design and Context

The study was carried out using a non-experimental design, with a descriptive methodology and cross-sectional inference.

The research was contextualized in the two Practicum subjects (I-initial or observation and II-final or intervention), which are included in the curricula of the bachelor's degrees in Early Childhood Education and Primary Education at the University of Salamanca [32]. Both degrees are taught at three different university centers, the University School of Education and Tourism in Ávila, the University School of Teaching in Zamora, and the Faculty of Education in Salamanca. With regard to the timing of the research, it took place during the 2020–2021 and 2021–2022 academic years, and it was developed in an exceptional situation caused by the COVID-19 pandemic (Table 2).

**Table 2.** Areas, competences and training objectives of the teacher training Practicum considered in the study.

| Context | Description |
|---|---|
| Degree. | |
| Early Childhood Education Teaching Degree. | University degree that enables students to exercise the regulated profession of Early Childhood Education Teacher, lasting 4 years (60 ECTS per year to complete 240) and structured in four modules: basic training in early childhood education (100 ECTS), didactic disciplinary in early childhood education (60 ECTS), electives and mentions (30 ECTS), external practices (44 ECTS) and final degree project (6 ECTS). |

**Table 2.** *Cont.*

| Context | Description |
|---|---|
| Childhood Education Teaching Degree | University degree that enables students to exercise the regulated profession of Primary Education Teacher, with a duration of 4 years (60 ECTS per year to complete 240) and structured in four modules: basic training in primary education (100 ECTS), disciplinary didactics in primary education (60 ECTS), optional subjects and mentions (30 ECTS), external internships (44 ECTS) and final degree project (6 ECTS). |
| Subject | |
| Practicum I | Subject with a load of 20 ECTS that is taught in the first semester of the third year and involves a 7-week practical period (200 h spent in a school). In general terms, the aim of Practicum I is for students to establish contact with a school, carrying out a global observation and reflection on the school and all the educational agents involved in it. Specific competences: 1. Trainees progressively assume their role as an educator in a school. 2. Identify the characteristics and functions of the educational professions. 3. Recognize the different elements that make up the school reality. 4. Collaborate with the school staff in order to achieve institutional objectives. 5. To perceive the processes of personal interaction and communication in the school. 6. To initiate a reflection, from real data, about the educational reality. 7. Acquire the necessary social skills to promote a favorable climate. 8. Appreciate the connotations of a collaborative relationship with families. 9. To present oneself as an authority in front of the students and maintain discipline. 10. Intervene appropriately in specific educational situations. |
| Practicum II | Subject with a load of 24 ECTS that is taught in the second semester of the fourth year and involves a 9-week practical period (240 h spent in the school). In general terms, the aim of Practicum II is for students to consolidate and develop the professional competences required on teaching degrees. Specific competences: 1. Participate in the professional activity of an educational center. 2. Identify different strategies for communication with families. 3. Initiate the collection of information about the family environment of the pupils. 4. Relate theory and practice in the management of the resources of a center. 5. Differentiate actions at the level of school, stage, cycle and classroom. 6. Collaborate with the educational community and the social environment. 7. Use appropriate techniques and strategies for the teaching activity. 8. To monitor the educational process of the students. |
| Academic Year | |
| 2020–2021 | The Ministry of Health agreed on coordinated actions against COVID-19 focused on the safe resumption of face-to-face educational activity. In this regard, the measures adopted for educational centers [17] included measures and recommendations at all centers and levels. The main measures and recommendations were as follows: limiting contacts, cleaning, disinfection and ventilation of centers, personal prevention measures, management and early detection of cases and action in the event of outbreaks according to protocols, prioritizing communication with families by telephone or email and carrying out administrative procedures telematically. As there were times when the epidemiological situation worsened in general or evolved differently in different territories, new documents and decrees were drawn up to provide a coordinated response to the transmission of COVID-19 [33]. In universities, the adapted face-to-face measures were of the same nature, with each university having to establish, in close collaboration with its competent educational administration, a contingency plan before the start of the academic year. These plans, based on the experience accumulated during the end of the 2019–2020 academic year, used digital forms of interaction in activities in which a face-to-face event or teaching could not take place. The latter had to be carried out with a distance of 1.5 m between students [18]. In this way, as exemplified by the University of Salamanca, the protocols of action for the different cases and circumstances, including in the Colegios Mayores, were published, providing the relevant protection material and resources that would help in their development [34]. |
| 2021–2022 | In the following academic year, prevention, hygiene and health-promotion measures against COVID-19 were similar in terms of trying to limit contacts, managing cases or adopting personal prevention measures, as well as cleaning and ventilation in centers. However, after the accumulated experience, the drafting of the document agreed with all the Autonomous Communities was more precise on some issues in relation to the 2020–21 academic year and took into account the ages of those potentially infected, other international recommendations, etc. [35]. Regarding the universities, the plan for academic adaptation was based on the health requirements imposed for this academic year. In this regard, the University of Salamanca itself informed all centers of the decision to adapt teaching to full attendance, following the good evolution of the epidemiological data and the high percentage of vaccination achieved, which led the Government of Castilla y León to declare a situation of "controlled risk" [36]. |

Source: The Authors.

## 2.2. Participants

The study population consisted of all students enrolled in the external practice subjects of the bachelor's degrees in Early Childhood Education and Primary Education at the University of Salamanca, in the academic years 2020–2021 and 2021–2022. The study sample comprised 725 students, aged between 20 and 42 years ($22.15 \pm 2.82$). Female participation was higher (80.6%), which is in line with the general trend in education

studies. The majority of students belonged to the Primary Education degree (65.2%), in line with the ratio of places on both degrees. Finally, the participation of students from the three schools was fairly equal. Table 3 shows the socio-demographic profile of the Practicum students in more detail.

**Table 3.** Socio-demographic profile of the students who participated in the study (*n* = 725).

| Variables | N (%) | Variables | N (%) |
|---|---|---|---|
| Gender | | Academic record (average grade) | |
| Female | 584 (80.6) | From 5 to 5.99 marks | 12 (1.7) |
| Male | 141 (19.4) | From 6 to 6.99 marks | 192 (26.5) |
| | | From 7 to 7.99 marks | 333 (45.9) |
| Age | | From 8 to 8.99 marks | 165 (22.8) |
| 20–22 years | 509 (70.2) | From 9 to 10 marks | 23 (3.2) |
| 23 or above | 216 (29.8) | | |
| | | Academic Year | |
| Campus location | | 2020–2021 | 413 (57.0) |
| ÁvilaCampus | 175 (24.1) | 2021–2022 | 312 (43.0) |
| Salamanca Campus | 230 (31.7) | | |
| Zamora Campus | 320 (44.1) | Subject | |
| | | Practicum I | 361 (49.8) |
| Degree | | Practicum II | 364 (50.2) |
| Early Childhood Education Teaching Degree | 252 (34.8) | | |
| Childhood Education Teaching Degree | 473 (65.2) | Type of center | |
| | | Public | 548 (75.6) |
| University major | | Private | 177 (24.4) |
| No Major | 319 (44.0) | | |
| Hearing and Speech | 56 (7.7) | Center location | |
| Special Education | 111 (15.3) | Urban | 541 (74.6) |
| Physical Education | 80 (11.0) | Rural | 184 (22.5) |
| Musical Training | 25 (3.4) | | |
| Realigious Education | 31 (4.3) | Preferred center/First option | |
| Foreign Language: German | 7 (1.0) | Yes | 614 (84.7) |
| Foreign Language: French | 18 (2.5) | No | 111 (15.3) |
| Foreign Language: English | 78 (10.8) | | |

*2.3. Instrument*

The data collection instrument was a questionnaire prepared ad hoc for the study and made up of four sections. The first section consisted of socio-demographic data collected through 11 items (gender, age, school, grade, year, record, subject, specialty, typology, tenure and preferred choice of school). In the second section, building on the work of Hamaidi et al. [13], eight items were presented on students' expectations regarding the achievement of the different formative objectives of the Practicum (e.g., develop my interaction skills, acquire classroom management skills, etc.). The same eight items were also presented to assess the students' perceptions of the negative impact of the COVID-19 pandemic on their achievement of the different Practicum learning objectives. The content and structure of the questionnaire were analyzed, with particular attention to the order and wording of questions. Special care was taken to avoid the introduction of biases inherent in self-administered questionnaires, such as social desirability bias, acquiescence bias, recall bias, and logic bias. Nevertheless, it is important to acknowledge that the chosen method of data collection represents a limitation of this research. The reliability index obtained for the scale as a whole was $\alpha = 0.865$. The third section included 10 items about the competences worked on during the Practicum, based on those selected in the theoretical framework, as well as asking about the influence of the pandemic on these competences. The reliability index obtained for the scale was $\alpha = 0.881$. In both sections, the response options were presented on a five-point Likert-type scale from 1 (strongly disagree)

to 5 (strongly agree). The reliability coefficient for the internal consistency of the entire questionnaire was satisfactory (Cronbach's α: 0.839).

### 2.4. Data Gathering

Prior to data collection, the Practicum coordinators of three university campuses were contacted in order to inform them about the study and obtain from the respondents their informed consent to participate. Data collection was carried out through voluntary participation by means of a self-administered questionnaire via email, with the link to the instrument attached, with guarantee of anonymity. It was elaborated using the Google Forms tool, selected on the basis of criteria of functionality and operability for the participants. The questionnaire was administered to students during their last week of work experience, situated at two different points in time: the beginning of February, in the case of Practicum I, and the end of April, for Practicum II. The questionnaires were completed voluntarily and anonymously, with prior express consent. To ensure data protection, the research process adhered to the ethical standards mandated by the University's Ethical committee, in accordance with both Spanish and European data protection regulations.

### 2.5. Data Analysis

Statistical analysis was performed with SPSS v28.0 (IBM, Armonk, NY, USA). Normality and homogeneity of the sample were examined using the Kolmogorov–Smirnov test. Descriptive statistics were calculated for continuous variables and qualitative variables to determine the relationship between the characteristics of the participants and the study variables. A Pearson correlation analysis was also performed to observe the relationship between factors. Differences between variables were analyzed using chi-squared tests (frequency distribution) and the Mann–Whitney U test for pairwise contrasts, given the lack of normality (comparison of means). Each item was used as a dependent variable, considering the sociodemographic variables of academic year, degree (training teacher) and subject (Practicum period) as grouping variables. The magnitudes of the differences or effect sizes were calculated using Cohen's d, interpreting the effects as null (0–0.2), low (0.20–0.50), moderate (0.50–0.79) or high (0.80). Significance was be set at the $p < 0.05$ level.

## 3. Results

### 3.1. Descriptive Analysis

In order to determine the students' perceptions of their expectations regarding the training objectives of the Practicum and the influence of COVID-19 on their achievement of these, the mean values obtained from the scores were used, considering the academic years, the degree program and the periods of the Practicum (Table 4).

Regarding the students' expectations in relation to the training objectives of the Practicum, for the total number of participants, the results showed that the best-rated competence was "(b) acquiring skills to manage/manage the classroom" (M = 4.57 SD = 0.72), followed by "(f) learning to prepare and plan teaching" (M = 4.53, SD = 0.80) and "(h) reaffirming my decision and vocation to be a teacher" (M = 4.57, SD = 0.72). On the other hand, the worst-rated was "(g) preparing educational software to support teaching" (M = 3.74, SD = 1.14). Regarding the influence of pandemic-related restrictions and security measures, overall, the students rated as the most negative aspects "(a) developing interaction and communication skills with students and families" (M = 2.96, SD = 1.23) and "(e) developing and diversifying teaching methods and strategies" (M = 2.62, SD = 1.31). On the other hand, the least influential aspect for the students was "(h) reaffirming the decision and vocation to be a teacher" (M = 1.85, SD = 1.30).

The disaggregated analysis, considering the academic year, degree and internship period, is shown in Figure 1. The results reveal that, in terms of expectations in teaching development, the mean scores were slightly higher for all the items in the academic year 2021–2022. This difference is also evident in the comparison of the influence of COVID-19 on these expectations, since this influence was significantly higher in the academic year 2020–2021. Considering now

the degree program, the mean scores for the Practicum expectations were slightly higher for all the items for the students on the Primary Education degree program. However, this trend does not correspond to the comparison of the influence of the pandemic on these expectations, since the students from the Early Childhood Education degree program stated that it had a greater influence on items c and d. In the comparison on the placement period, the mean scores for the Practicum expectations were similar for all the items. On the other hand, in the comparison of the influence of COVID-19 on these expectations, the students in the first period rated the influence of the pandemic higher for all the items.

**Table 4.** Descriptive data on the expectations of the Practicum and the influence of COVID-19 according to academic year, degree and subject.

| Item (Objectives) | | Academic Year | | | | Degree | | | | Subject | | | |
|---|---|---|---|---|---|---|---|---|---|---|---|---|---|
| | | 2020–2021 | | 2021–2022 | | Early Childhood Education | | Childhood Education | | PI | | PII | |
| | | M | SD | M | SD | M | SD | M | SD | M | SD | M | SD |
| (a) Develop my interaction and communication skills with pupils and families | Expectations | 4.38 | 0.831 | 4.45 | 0.796 | 4.33 | 0.904 | 4.45 | 0.764 | 4.40 | 0.817 | 4.42 | 0.817 |
| | Influence | 3.12 | 1.209 | 2.76 | 1.218 | 2.79 | 1.268 | 3.06 | 1.193 | 3.07 | 1.189 | 2.85 | 1.252 |
| (b) Acquire skills to manage/manage the classroom | Expectations | 4.55 | 0.727 | 4.59 | 0.712 | 4.50 | 0.796 | 4.60 | 0.675 | 4.59 | 0.698 | 4.55 | 0.743 |
| | Influence | 2.49 | 1.228 | 2.23 | 1.223 | 2.26 | 1.222 | 2.44 | 1.234 | 2.55 | 1.224 | 2.21 | 1.219 |
| (c) Increase my knowledge of the functioning and management of the school | Expectations | 4.36 | 0.886 | 4.45 | 0.835 | 4.30 | 0.959 | 4.45 | 0.807 | 4.40 | 0.835 | 4.39 | 0.895 |
| | Influence | 2.48 | 1.312 | 2.13 | 1.217 | 2.38 | 1.296 | 2.30 | 1.277 | 2.38 | 1.279 | 2.28 | 1.287 |
| (d) Develop my communication and cooperation skills with my fellow teachers | Expectations | 4.17 | 0.974 | 4.23 | 0.971 | 4.09 | 1.037 | 4.26 | 0.932 | 4.14 | 0.998 | 4.26 | 0.944 |
| | Influence | 2.65 | 1.272 | 2.28 | 1.207 | 2.54 | 1.276 | 2.46 | 1.247 | 2.57 | 1.263 | 2.41 | 1.247 |
| (e) Develop and diversify my teaching methods and strategies | Expectations | 4.46 | 0.816 | 4.48 | 0.802 | 4.35 | 0.896 | 4.53 | 0.753 | 4.47 | 0.799 | 4.47 | 0.821 |
| | Influence | 2.80 | 1.294 | 2.39 | 1.291 | 2.38 | 1.265 | 2.75 | 1.314 | 2.71 | 1.322 | 2.54 | 1.29 |
| (f) Learn how to plan and arrange teaching | Expectations | 4.52 | 0.811 | 4.54 | 0.785 | 4.42 | 0.905 | 4.58 | 0.732 | 4.53 | 0.782 | 4.53 | 0.818 |
| | Influence | 2.49 | 1.271 | 2.15 | 1.207 | 2.17 | 1.217 | 2.44 | 1.266 | 2.39 | 1.224 | 2.30 | 1.284 |
| (g) Prepare educational software to support teaching | Expectations | 3.71 | 1.165 | 3.77 | 1.098 | 3.50 | 1.219 | 3.86 | 1.071 | 3.67 | 1.13 | 3.8 | 1.141 |
| | Influence | 2.28 | 1.156 | 1.99 | 1.118 | 2.05 | 1.123 | 2.21 | 1.159 | 2.20 | 1.149 | 2.11 | 1.148 |
| (h) Reaffirm my decision and vocation to become a teacher | Expectations | 4.50 | 0.949 | 4.57 | 0.861 | 4.53 | 0.908 | 4.53 | 0.916 | 4.56 | 0.89 | 4.50 | 0.934 |
| | Influence | 1.93 | 1.338 | 1.75 | 1.233 | 1.83 | 1.291 | 1.87 | 1.30 | 1.88 | 1.348 | 1.83 | 1.244 |

In order to determine the students' perceptions of the competences they developed during their placements, we used the mean values obtained from the scores, considering the academic years, the degree programs and the placement periods (Table 5).

With regard to the competences developed in the Practicum, for the total number of participants, the results showed that the highest-rated competence was "1. Acquire classroom management skills" (M = 4.58, SD = 0.64), followed by "2. Develop social and communicative skills to turn the classroom into a place of learning and coexistence" (M = 4.50, SD = 0.70) and "5. Identify my functions as a teacher and develop them" (M = 4.50, SD = 0.72). On the other hand, the worst-rated was "9. Knowing and putting into practice contact and relations with families" (M = 3.29, SD = 1.39) and "8. Knowing ways of collaborating with the different sectors of the educational community and the social environment" (M = 3.83, SD = 1.13).

The disaggregated analysis, considering the academic year, degree and internship period, is shown in Figure 2. Regarding the academic year, the students' scores were better for all the items except item 10 in the 2021–2022 academic year. With regard to degree and internship, the scores were similar in both comparisons.

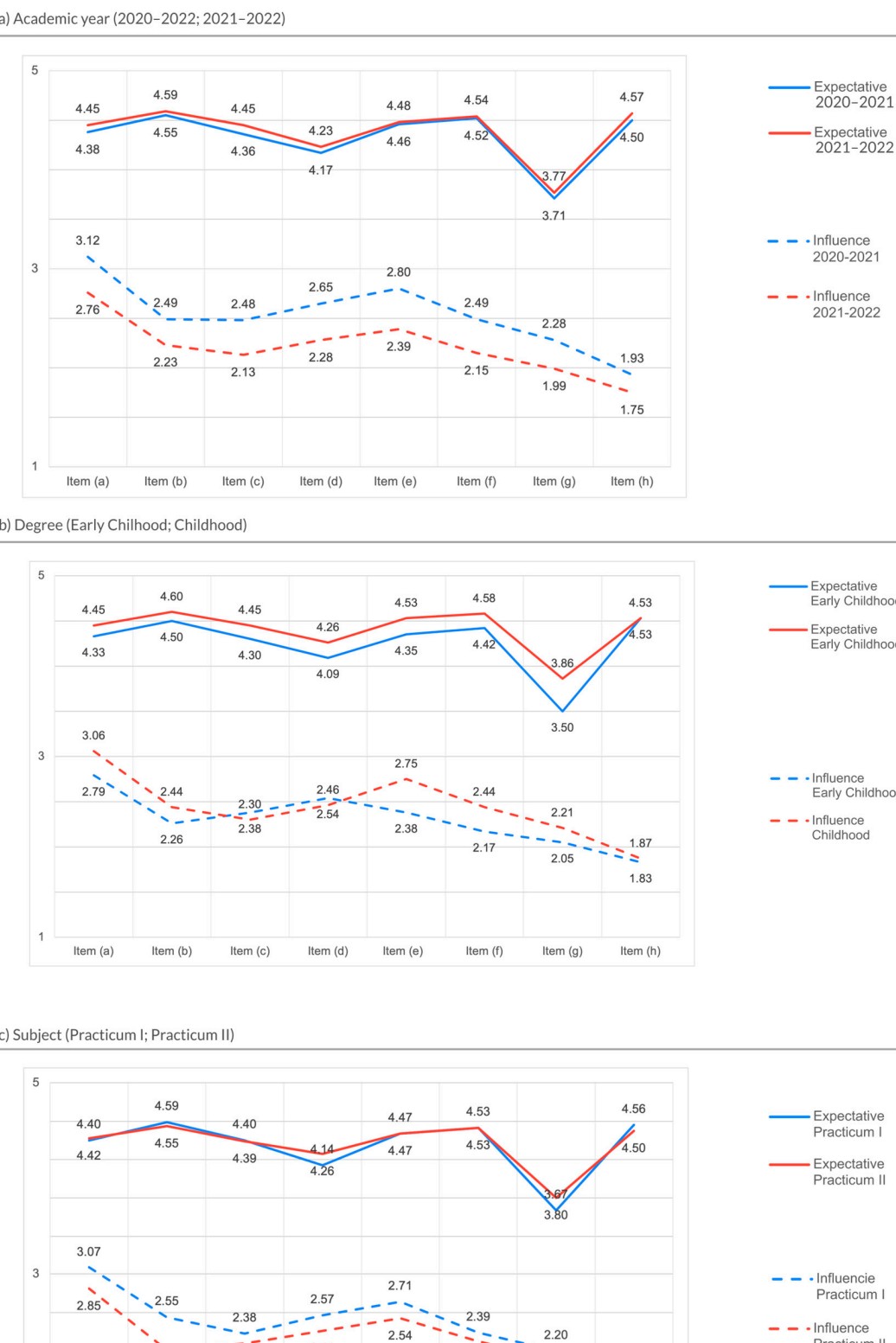

**Figure 1.** Expectations towards the Practicum and the influence of COVID-19 according to academic year, degree and subject.

**Table 5.** Descriptions of the competences developed in the Practicum according to academic year, degree and subject.

| Item (Competences) | Academic Year | | | | Degree | | | | Subject | | | |
|---|---|---|---|---|---|---|---|---|---|---|---|---|
| | 2020–2021 | | 2021–2022 | | Early Childhood Education | | Childhood Education | | PI | | PII | |
| | M | SD | M | SD | M | SD | M | SD | M | SD | M | SD |
| (1) Acquire practical knowledge of the classroom and classroom management | 4.53 | 0.659 | 4.65 | 0.597 | 4.59 | 0.621 | 4.57 | 0.644 | 4.59 | 0.585 | 4.57 | 0.683 |
| (2) Develop social and communicative skills to turn the classroom space into a place of learning and coexistence | 4.42 | 0.735 | 4.61 | 0.621 | 4.48 | 0.7 | 4.51 | 0.692 | 4.48 | 0.687 | 4.52 | 0.702 |
| (3) Learn and use different didactic strategies for the development of teaching-and-learning processes | 4.34 | 0.808 | 4.51 | 0.735 | 4.46 | 0.688 | 4.39 | 0.826 | 4.39 | 0.771 | 4.44 | 0.792 |
| (4) Relate the theoretical and practical concepts addressed in the different subjects of the degree with the reality of a classroom and educational center | 3.95 | 1.037 | 4.16 | 0.971 | 4.13 | 0.967 | 3.99 | 1.035 | 4.06 | 0.976 | 4.01 | 1.051 |
| (5) Identify teachers' roles and develop them | 4.45 | 0.731 | 4.57 | 0.691 | 4.49 | 0.749 | 4.51 | 0.698 | 4.48 | 0.703 | 4.52 | 0.729 |
| (6) Participate in the improvement proposals and the different activities proposed by an educational center, beyond the content teaching in the different areas that can be established in a center | 4.07 | 1.067 | 4.36 | 0.875 | 4.08 | 1.085 | 4.26 | 0.946 | 4.1 | 1.031 | 4.29 | 0.958 |
| (7) To regulate the processes of interaction and communication in groups of students aged 6–12 years (Primary Education/Childhood Education) or 3–6 years (Early Childhood/Infant Education) | 4.06 | 1.021 | 4.36 | 0.833 | 4.12 | 1.044 | 4.22 | 0.905 | 4.14 | 0.928 | 4.24 | 0.981 |
| (8) Know ways of collaborating with the different sectors of the educational community and the social environment | 3.63 | 1.198 | 4.09 | 0.979 | 3.73 | 1.19 | 3.89 | 1.097 | 3.75 | 1.141 | 3.92 | 1.118 |
| (9) Understand and establish contact and relationships with families | 3.08 | 1.372 | 3.57 | 1.364 | 3.37 | 1.349 | 3.25 | 1.409 | 3.24 | 1.361 | 3.34 | 1.416 |
| (10) Program, direct, execute and assess, with the appropriate supervision, a Teaching Unit and the student activities the teacher-tutor considers appropriate | 4.19 | 0.998 | 4.14 | 1.09 | 4.11 | 1.073 | 4.2 | 1.019 | 4.01 | 1.079 | 4.33 | 0.974 |

### 3.2. Inferential Analysis

In order to explore significant differences between the students' perceptions of their expectations of the Practicum and the influence of the pandemic's consequences, an inferential analysis was carried out according to academic year, degree and placement period (Table 6).

With regard to the students' expectations of the training objectives of the Practicum, no significant differences were found in the comparison by academic year or Practicum period. With regard to the comparison by degree, significant values were obtained for objectives (d), (e) and (g), with the average rank of the Primary Education degree being higher than that of Early Childhood Education in all three. With regard to the scores for the influence of COVID-19 on the Practicum training, the results reveal significant differences. Regarding the students' scores on the influence of COVID-19, the results reveal significant differences for all the items except objective (h) between the two academic years, with the mean rank of the 2021–2022 academic year being lower than that of the 2020–2021 academic year for all of the items. Significant values were also obtained in the comparison by degree (expectations a, b, e and f), and by internship (in a and b). For all of the items, a null effect size was found (<0.20).

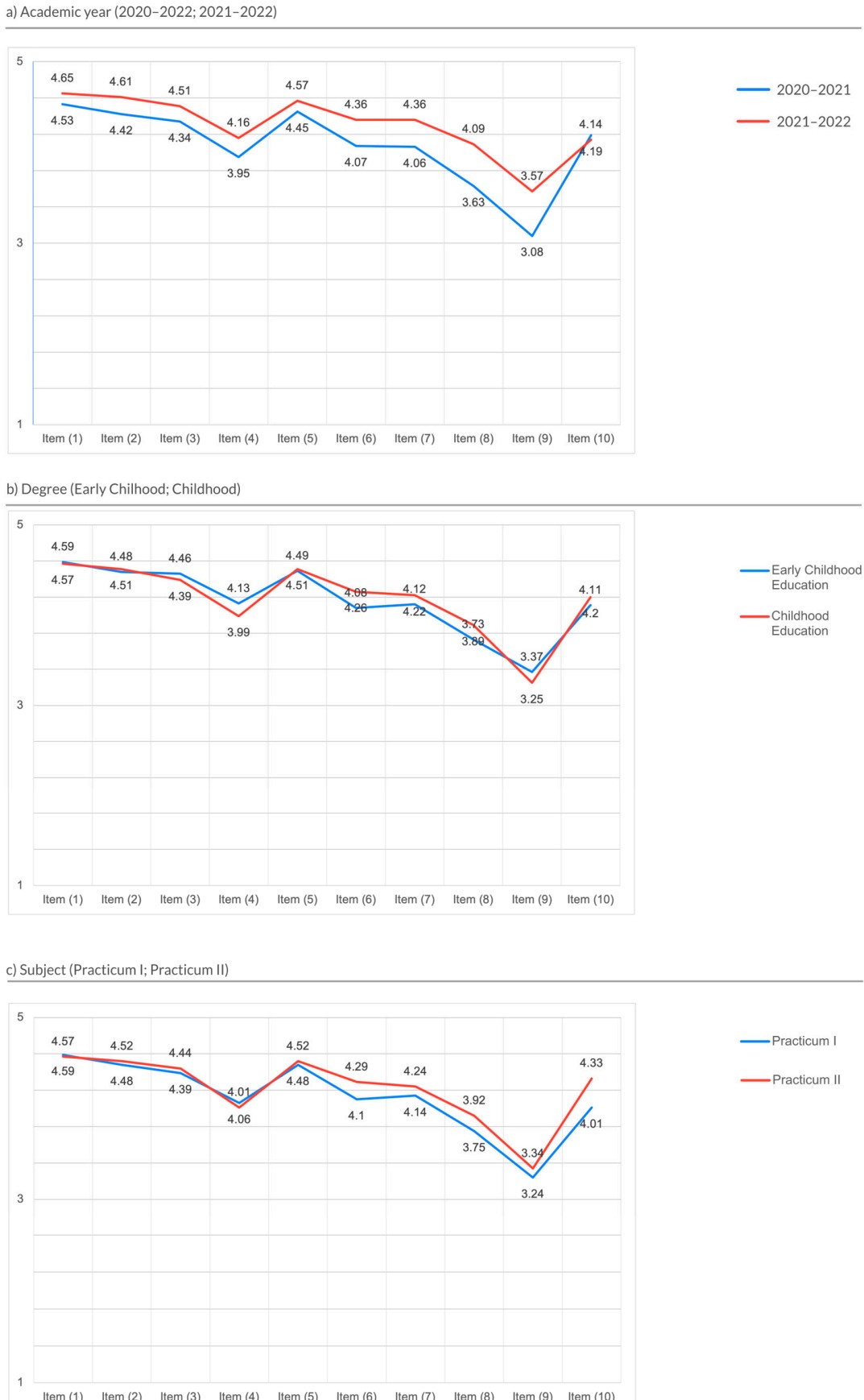

**Figure 2.** Competences developed in the Practicum according to academic year, degree and subject.

Similarly, in order to explore the significant differences between the students' perceptions of the competences they developed during the Practicum, an inferential analysis was carried out according to academic year, degree and Practicum period (Table 7). The results reveal significant differences between the academic years 2020–2021 and 2021–2022 for all the items except competence 10. Significant values were also obtained for items 6, 7, 8 and 10 between the practical periods.

**Table 6.** Inferential analysis of the expectations of the Practicum and the influence of COVID-19 according to academic year, degree and subject.

| Item (Objectives) | | Academic Year | | | Degree | | | Subject | | |
|---|---|---|---|---|---|---|---|---|---|---|
| | | z | p | r | z | p | R | z | p | r |
| (a) Develop my interaction and communication skills with pupils and families | Expectations | 1.380 | 0.168 | 0.05 | 1.310 | 0.190 | 0.05 | 0.456 | 0.648 | 0.02 |
| | Influence | −3.964 | <0.001 ** | 0.15 | 2.828 | 0.005 * | 0.11 | −2.402 | 0.016 * | 0.09 |
| (b) Acquire skills to manage/manage the classroom | Expectations | 0.663 | 0.507 | 0.02 | 1.140 | 0.254 | 0.04 | −0.572 | 0.567 | 0.02 |
| | Influence | −3.002 | 0.003 | 0.11 | 1.961 | 0.050 * | 0.07 | −3.833 | <0.001 ** | 0.14 |
| (c) Increase my knowledge of the functioning and management of the school | Expectations | 1.242 | 0.214 | 0.05 | 1.707 | 0.088 | 0.06 | 0.333 | 0.739 | 0.01 |
| | Influence | −3.714 | <0.001 ** | 0.14 | −0.838 | 0.402 | 0.03 | −1.244 | 0.213 | 0.05 |
| (d) Develop my communication and cooperation skills with my fellow teachers | Expectations | 1.012 | 0.312 | 0.04 | 2.035 | 0.042 * | 0.08 | 1.731 | 0.083 | 0.06 |
| | Influence | −3.877 | <0.001 ** | 0.14 | −0.844 | 0.399 | 0.03 | −1.681 | 0.093 | 0.06 |
| (e) Develop and diversify my teaching methods and strategies | Expectations | 0.520 | 0.603 | 0.02 | 2.447 | 0.014 * | 0.09 | 0.511 | 0.609 | 0.02 |
| | Influence | −4.336 | <0.001 ** | 0.16 | 3.577 | <0.001 ** | 0.13 | −1.663 | 0.096 | 0.06 |
| (f) Learn how to plan and arrange teaching | Expectations | 0.372 | 0.710 | 0.01 | 1.888 | 0.059 | 0.07 | 0.409 | 0.683 | 0.02 |
| | Influence | −3.766 | <0.001 ** | 0.14 | 2.799 | 0.005 * | 0.10 | −1.190 | 0.234 | 0.04 |
| (g) Prepare educational software to support teaching | Expectations | 0.586 | 0.558 | 0.02 | 3.685 | <0.001 ** | 0.14 | 1.516 | 0.129 | 0.06 |
| | Influence | −3.544 | <0.001 ** | 0.13 | 1.799 | 0.072 | 0.07 | −1.080 | 0.280 | 0.04 |
| (h) Reaffirm my decision and vocation to become a teacher | Expectations | 0.939 | 0.348 | 0.03 | −0.416 | 0.678 | 0.02 | −0.567 | 0.571 | 0.02 |
| | Influence | −1.920 | 0.55 | 0.07 | 0.413 | 0.680 | 0.02 | −0.106 | 0.915 | 0.00 |

\* $p < 0.05$; \*\* $p < 0.01$.

**Table 7.** Inferential analysis of the competences developed in the Practicum according to academic year, degree and subject.

| Item (Competences) | Academic Year | | | Degree | | | Subject | | |
|---|---|---|---|---|---|---|---|---|---|
| | Z | p | r | z | p | R | z | p | r |
| (1) Acquire practical knowledge of the classroom and classroom management | −2.781 | 0.005 * | 0.10 | −0.373 | 0.709 | 0.01 | 0.369 | 0.712 | 0.01 |
| (2) Develop social and communicative skills to turn the classroom space into a place of learning and coexistence | −3.616 | <0.001 ** | 0.13 | 0.620 | 0.536 | 0.02 | 1.243 | 0.214 | 0.05 |
| (3) Learn and use different didactic strategies for the development of teaching-and-learning processes | −2.919 | 0.004 * | 0.11 | −0.548 | 0.584 | 0.02 | 1.204 | 0.229 | 0.04 |
| (4) Relate the theoretical and practical concepts addressed in the different subjects of the degree with the reality of a classroom and educational center | −2.806 | 0.005 * | 0.10 | −1.729 | 0.084 | 0.06 | −0.292 | 0.770 | 0.01 |
| (5) Identify teachers' roles and develop them | −2.607 | 0.009 * | 0.10 | −0.061 | 0.951 | 0.00 | 1.391 | 0.164 | 0.05 |
| (6) Participate in the improvement proposals and the different activities proposed by an educational center, beyond the content teaching in the different areas that can be established in a center | −3.561 | <0.001 ** | 0.13 | 1.816 | 0.069 | 0.07 | 2.893 | 0.004 * | 0.11 |
| (7) To regulate the processes of interaction and communication in groups of students aged 6–12 years (Primary Education/Childhood Education) or 3–6 years (Early Childhood/Infant Education) | −3.977 | <0.001 ** | 0.15 | 0.866 | 0.387 | 0.03 | 2.190 | 0.029 * | 0.08 |
| (8) Know ways of collaborating with the different sectors of the educational community and the social environment | −5.149 | <0.001 ** | 0.19 | 1.547 | 0.122 | 0.06 | 2.161 | 0.031 * | 0.08 |
| (9) Understand and establish contact and relationships with families | −4.746 | <0.001 ** | 0.18 | −1.033 | 0.301 | 0.04 | 1.016 | 0.310 | 0.04 |
| (10) Program, direct, execute and assess, with the appropriate supervision, a Teaching Unit and the student activities the teacher-tutor considers appropriate | −0.209 | 0.835 | 0.01 | 1.111 | 0.266 | 0.04 | 4.535 | <0.001 ** | 0.17 |

\* $p < 0.05$; \*\* $p < 0.01$.

## 4. Discussion

After analyzing the data collected, in this study, we can highlight some issues that open a space for reflection and connection with observations in other studies. Firstly, after the number of items assessed among the expectations and competences worked on in the Practicum, the importance of the subject in teacher training and the need to continue to study in depth the previous expectations of future teachers seem to have been evidenced, as in other studies [10]. University training is mainly theoretical and the achievement of competences requires a context of performance and practical application, such as in the Practicum, which manages to overcome the gap that still exists between the ways of learning in real professional contexts and that which often occurs in academic contexts, despite the recurrent call for coordinating actions between the two contexts, theorical and practical [1,4,37–39].

From the analysis of the students' expectations, two relevant aspects for teacher training can be highlighted. Firstly, that the students' learning expectations in the professional context focus on the second of the areas mentioned (competences for managing coexistence and participation in the classroom), because, when asked about the main limitations imposed by the pandemic, the participants in this study stated that their interaction and communication skills could not be fully developed because of the pandemic, as other studies confirmed [20,23]. Secondly, pedagogical and didactic competences, which the students would be expected to gain relatively rapidly, were in second place, suggesting that internship periods should prepare trainees to know how to plan teaching. Furthermore, that the main limitations were associated with the impossibility of diversifying teaching methods because of the social restrictions that impeded interaction, an issue that was not highlighted by previous. The third area, competences for collaborative work and professional development, was not particularly prominent among the students' expectations, although it is clear from their answers that the limitations due to the pandemic did not influence their reaffirmation of their commitment to the profession, and that they accepted that it was an exceptional situation. This progression towards a more positive view of their learning expectations was verified in the comparative analysis by academic year or by typology, which demonstrated that at the beginning of the pandemic, the trainees' expectations were lower and as the evolution was positive, so were their expectations. Interestingly, by degree, the students on the Early Childhood Education Teaching degree expressed lower learning expectations than the students on the Primary Education Teaching degree, although they had higher expectations related to the third area, competences for collaborative work and professional development.

In relation to the ten competences that were analyzed, the students seem to have considered that they were more prepared by their university training with regard to the pedagogical and didactic competences (the first area) because these were not the aspects that they valued as being most developed during their work placements. However, they considered that they had learnt the most in the competences of area 2, the competences of managing coexistence and participation, as well as those of area 3, especially knowing how to identify their functions as teachers in the classroom reality and knowing how to collaborate with the different agents of the educational community. These results are relevant for the future, for the revision of teacher training curricula, through which the training of students in these aspects of improving coexistence and the creation of teacher-collaboration networks could be reinforced.

In short, the importance for trainee teachers of learning related to the reality of the classroom and the school environment cannot be ignored, because it is a major factor for the growth of teachers who are beginning their professional development [40,41]. Thus, it is essential to continue working on competences and areas related to professional development through training at universities, so that internships are not students' first encounters with reality. Similarly, other studies confirmed the use of case studies or other methods of approaching classroom situations and interactions, such as by presenting new conflicts and existing circumstances, so that undergraduate students can undergo training

in practical strategies, considering that in the future, further tools and knowledge for prevention will be needed [42].

Another aspect that was also shown to be lacking during the practice processes that took place during the pandemic was the possibility of knowing and understanding what happens in the school context, which also extends to families, as other studies have shown [43]. Thus, it is worth highlighting the students' assessment of the problems caused by the pandemic in order to develop interaction and communication skills with students and families. Similar studies with Practicum students have identified the relationship between the school and the family as fundamental, and it is necessary to reflect on how to reinforce and include this learning around tutorial action and new channels of communication between these stakeholders [43].

The impact of digitalization, which was forced upon many didactic and pedagogical interactions during the pandemic, is barely noticeable in the results. Although the teaching-and-learning process was carried out in virtual environments during the period of confinement, the reality unleashed by the pandemic was highly challenging for higher education. This was especially evident in terms of the ability of the higher education system to respond to the common hurdles with which it had dealt with for years: the fleeting era of digitalization and the sometimes problematic and dehumanizing communication through screens [44]. In this study and in the context of the limitations of the pandemic in schools, the Practicum students did not emphasize that they were going to learn about digital teaching methods, given that in our country, distancing measures were implemented to ensure safety. At the same time, this revealed shortcomings in learning that were affected by the limitations on interaction.

In this sense, the evidence for the negative influence of isolation during the pandemic was clear, including the evidence in our study, in which the students on both degrees perceived more or less equal differences between the more restrictive first year 2020–2021 and the following year. Thus, the prioritization of direct contact between the different actors in the educational community and its effect on decreasing contagion made clear the importance of education as interaction. This is a concern that was also highlighted in some of the other studies mentioned above [20,23]. Further research is needed to reaffirm these or other ideas in post-pandemic contexts to establish whether the pandemic context and the improvement that occurred, for example, through the renewed contact between different educational agents, has benefited professional training

## 5. Conclusions

This study contributes to incorporating elements of reflection into the analysis of academic competences and to the study of their relationship with other, non-academic gains acquired through the trainee teachers' Practicum. The study emphasized the significance of students' perceptions and expectations about the learning acquired in the Practicum, since if of competence gains in certain areas are expected, there will be a tendency to perceive the acquisition of competences in these areas, even if the Practicum process takes place in unusual circumstances, such as the pandemic. The results can also be used for future comparative studies to verify whether students from other universities, national or international, perceive gains in the pedagogical and didactic aspects in a similar manner those observed in this research or in other areas. In addition, this study also helps to understand the evolution of teacher training plans and to understand how prepared students perceive themselves to be in the three areas of competence to develop their Practicum, which is the prelude to their professional reality.

**Author Contributions:** Authors contributed equally to this work. All authors have read and agreed to the published version of the manuscript.

**Funding:** This research received no external funding.

**Institutional Review Board Statement:** After consultation with the Research Ethics Committee of the University of Salamanca, the committee determined that the evaluation was not necessary, as none of the research questions could identify the participants, guaranteeing their anonymity.

**Informed Consent Statement:** Informed consent was obtained from all subjects involved in the study.

**Data Availability Statement:** The datasets generated and analyzed during the current study are available from the corresponding author on reasonable request.

**Acknowledgments:** The study was developed under the framework of the project: Banco de casos prácticos basados en el Modelo de Competencias Profesionales del Docente de Castilla y.

**Conflicts of Interest:** The authors declare no conflict of interest.

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
