# Peer review of "Competences Expected and Gained during the Teaching Practicum: Analysis of Three Competence Areas Affected during the Pandemic"

_education, doi:10.3390/educsci14010088_

Round 1

Reviewer 1 Report

Comments and Suggestions for Authors

Thank you for the opportunity to review your paper. It is a topic that impacted many people in different ways in teacher education, how to best support the practicum experience, and learning more about what student teachers' experiences were while on practicum is an important aspect to consider.

I have gone through the attached paper and made some minor comments throughout. It was important to clarify that the study had gone through the process of a university-led ethics process, as this was not made clear in the paper. 

I wish you well in publishing this paper.

Comments on the Quality of English Language

There were some minor edits to be made - please see the PDF. 

Author Response

Thank you very much for taking the time to review this manuscript and for corrections.

About your comments: 

  1. English language has been revised and some minor editing has been carried out.
  2. It was clarified that the study had gone through the process of a university-led ethics process. 

Thanks you again.

Reviewer 2 Report

Comments and Suggestions for Authors

The tables should summarize information. Also, use the text to introduce the tables.

  1. What is the main question addressed by the research? Basically, this is a survey study on student outcomes during the pandemic. Although the authors state that they are extending prior studies, there needs to be explicit discussion on how the study’s findings expand or contradict with the existing literature. Also, in the introduction, there needs to a summary of prior studies.
  2. Do you consider the topic original or relevant in the field? Does it
    address a specific gap in the field?
    This is difficult to assess since there is no mention of prior studies. Lines 75-96 can be summarized instead since it looks like a copy and paste of the competences somewhere. I recommend summarizing this part while integrating with prior studies and adding the contributions to the field.
  3. What does it add to the subject area compared with other published
    material?
    Same as point 2
  4. What specific improvements should the authors consider regarding the
    methodology? What further controls should be considered?
    This area seems detailed enough, but there is no mention if the survey was validated.
  5.   Are the conclusions consistent with the evidence and arguments presented
    and do they address the main question posed?
    Although the authors present the findings in the discussion section, there needs to a discussion section dedicated to articulating how the results help form policies or improvements to practicum experiences.
  6. Are the references appropriate? Yes
  7. Please include any additional comments on the tables and figures. Shortening the text in the table is needed here. Also, authors need to articulate the contents of tables. Figures are blurry or low-quality resolution.

Comments on the Quality of English Language

The tables are very dense. These tables should be a summary of the text. Also proper introductions to such tables need to be articulated more.

Author Response

Thank you very much for taking the time to review this manuscript and for corrections. 

About your comments:

  1. The paragraphs in the introductory section have been reordered.
  2. A summary of previous studies has been included in the introduction section.
  3. Lines 75-96 of the introductory section have been omitted.
  4. The discussion and conclusion sections have been separated.
  5. The figures have been edited to improve resolution. 
  6. Regarding the tables, we consider that the current format facilitates the understanding of the context. However, if the editor considers it necessary, table 2 can be converted into a paragraph and in tables 4-7 the objectives and competences can be coded.

Thank you again.

Round 2

Reviewer 2 Report

Comments and Suggestions for Authors

Thank you for the detailed responses to my review and the changes in the manuscript. A minor item would be adding the number of items used in the questionnaire in line 163. Then, add the number of items used in each instrument section as you did in sections 2 (8 items) and 3 (10 items). Since it is a self-administered questionnaire, there needs to be a limitation related to social desirability bias somewhere in the methods section.

Author Response

Thank you for your comments.

We have modified the methods section in response to your suggestions.

  1. We add the number of elements used in each instrument section (line 163-169
  2. We add a paragraph related to biases inherent in self-administered questionnaires (lines 171-176).

    The content and structure of the questionnaire were analyzed, with particular attention to the order and wording of questions. Special care was taken to avoid the introduction of biases inherent in self-administered questionnaires, such as social desirability bias, acquiescence bias, recall bias, and logic bias. Nevertheless, it is important to acknowledge that the chosen method of data collection represents a limitation in this research.